# Extracellular Vesicles miRNA Cargo for Microglia Polarization in Traumatic Brain Injury

**DOI:** 10.3390/biom10060901

**Published:** 2020-06-12

**Authors:** Maria Antonietta Panaro, Tarek Benameur, Chiara Porro

**Affiliations:** 1Department of Biosciences, Biotechnologies and Biopharmaceutics, University of Bari, 70125 Bari, Italy; mariaantonietta.panaro@uniba.it; 2Department of Biomedical Sciences, College of Medicine, King Faisal University, 31982 Al-Ahsa, Kingdom of Saudi Arabia; tbenameur@kfu.edu.sa; 3Department of Clinical and Experimental Medicine, University of Foggia, 71122 Foggia, Italy

**Keywords:** TBI, neuroinflammation, microglia, Extracellular Vesicles, Exosomes, miRNAs

## Abstract

Traumatic brain injury (TBI) is one of the major causes of death and disability worldwide, and despite its high dissemination, effective pharmacotherapies are lacking. TBI can be divided into two phases: the instantaneous primary mechanical injury, which occurs at the moment of insult, and the delayed secondary injury, which involves a cascade of biological processes that lead to neuroinflammation. Neuroinflammation is a hallmark of both acute and chronic TBI, and it is considered to be one of the major determinants of the outcome and progression of disease. In TBI one of the emerging mechanisms for cell–cell communication involved in the immune response regulation is represented by Extracellular Vesicles (EVs). These latter are produced by all cell types and are considered a fingerprint of their generating cells. Exosomes are the most studied nanosized vesicles and can carry a variety of molecular constituents of their cell of origin, including microRNAs (miRNAs). Several miRNAs have been shown to target key neuropathophysiological pathways involved in TBI. The focus of this review is to analyze exosomes and their miRNA cargo to modulate TBI neuroinflammation providing new strategies for prevent long-term progression of disease.

## 1. Introduction

Traumatic brain injury (TBI) is a serious public health problem and, at the moment, no effective clinical treatment strategies have been found. TBI indicates a sudden trauma, induced by vehicle collisions, wars, violence, terrorism, falls, or sporting activity [1]. TBI is a multiphase pathology with complex interactions between brain, periphery and the immune system [2].

TBI-associated brain damage has two phases: (a) the first phase takes place when the insult happens, and includes diffuse axonal injury, contusion and laceration, and intracranial hemorrhage that can result in instantaneous cell death; (b) the second phase involves a cascades of biological processes that lead to neuroinflammation, damage of the blood–brain barrier (BBB) allowing infiltration of peripheral immune cells into the brain and aggravation of neuroinflammatory response, oxidative stress, and apoptosis. The immediate primary injury is considered untreatable; rather, the second phase of injury has received more attention because some therapeutic interventions could be carried out [3,4,5,6,7].

In neuroinflammation following TBI, the activation of the resident glia (microglia and astrocytes), release of inflammatory mediators in the brain, and recruitment of immune cells have been described [8].

During neuroinflammation, microglia constitute the first line of defense whenever damage has occurred [9], producing both pro- and anti-inflammatory mediators, with the latter being indispensable for promoting neuroprotection. However, if the production of pro-inflammatory mediators is excessive and prolonged, brain damage may be exacerbated [9]. 

Astrocytes are a very abundant cell type in the central nervous system (CNS) and are very important in maintaining the stability of microenviroment. Astrocytes and microglia play a critical role in TBI. In particular, activated astrocytes produce different factors that may influence microglia [10,11].

Extracellular vesicles (EVs) are considered a new important way of facilitating intercellular communication.

The family of EVs is composed of heterogenous mixture of exosomes (Exos, 10–100 nm diameter), microvesicles (MVs, 100–1000 nm), and apoptotic bodies (1–5 nm). All cell types produce EVs, and both EV content and number change dynamically in response to specific external signals. EVs consist of a lipid bilayer membrane and contain parental cell-derived active cargos such as lipids, proteins, DNAs, mRNAs, microRNAs (miRNAs), non-coding RNAs, and organelles [12]. 

The role of EVs in the brain has gained considerable attention in recent years. Of particular interest, EVs in the brain play a vital role in modulating the synaptic activity and neuronal communication [13,14], contributing to the pathogenesis of several neurodegenerative diseases, such as multiple sclerosis [15], Alzheimer’s disease (AD) [16], prion disease [17] and Huntington’s disease [18].

Among EVs, the action of exosomes in CNS is very studied especially their miRNA cargo. The current review summarizes the role of EVs in CNS with particular focus on EVs that are originated from or are delivered to microglia, their cargo and the potential therapeutic applications of EVs.

## 2. Microglia: Neuroinflammation and TBI

Microglia, the resident immune cells of the brain, are involved in both essential physiological functions and in pathological conditions of the CNS [19,20,21]. Their true origin has been the subject of debate, since previous evidence suggested that microglia differentiate in the bone marrow from embryonic hematopoietic precursor cells. However, more recent studies in mice have reported that these cells are derived from progenitors in the embryonic yolk sac early during development, entering the CNS through the blood vasculature before closure of the BBB [22]. Microglia play a crucial role in refining neural circuits in brain development, regulating synaptic networks, promoting synaptic formation and maturation, through synaptic pruning, neuronal apoptosis, neurogenesis and secretion of growth factors [23,24,25,26].

As well as macrophages, microglia have a double reactive activity, given the existence of two different phenotypes: the classic activated state (M1) associated with pro-inflammatory functions and the alternative state (M2) associated with anti-inflammatory properties, although in vivo this functional profile does not seem to be strictly applicable [27].

In this regard, pro-inflammatory polarization is characterized by the release of several pro-inflammatory mediators (TNF-α, IL-1β, CD86, CD16/CD32 and iNOS), while anti-inflammatory polarization is discernible by the expression of anti-inflammatory factors (IL-4, IL-10, Arg1, IGF1 and CD206) [28].

Our previous studies have reported that modulating pro-inflammatory polarization consenting the shifting towards anti-inflammatory polarization improved neuroinflammatory pictures in vivo [28]. In addition, since in pro-inflammatory polarization the TLR4/NF-κB signaling pathway is significantly activated, pharmacologically targeting this pathway, an attenuation of the inflammatory response was observed [29,30,31,32,33].

An interesting aspect regarding the gene expression profile in correspondence with their activated state is that the microglial phenotype is triggered by various factors, and this shifting can also occur during disease progression [34,35,36].

Indeed, it has been described that early microglia activation following TBI may contribute to the restoration of homeostasis in the brain.

TBI is an injury of the brain structure due to an extrinsic biomechanical insult to the cranium, evolving in neuronal, axonal and vascular damage. Within minutes of TBI there is a robust neuroinflammatory response that is mediated by complex molecular and cellular events orchestrating a complex immunological tissue reaction like ischemic reperfusion injury [37]. Microglial activation occurs early after experimental mice and human TBI [38,39,40], and can persist for years, detectable both in vivo and post-mortem. Secondary injury follows the primary injury-mediated neuroinflammation may persist from hours to days after the initial insults. Moreover, neuroinflammation is followed by secondary injury that can hinder the neuroprotection and reparative mechanisms linked to delays and limited neurobehavioral recovery after TBI [41]. In this regard, prolonged microglial activation is detrimental to the CNS as they produce nitric oxide and reactive oxygen intermediates other than pro-inflammatory cytokines causing neuronal dysfunction and death. Sites of microglia activation often coincide with neuronal degeneration and axonal damage [42,43], thus leading to the hypothesis that the inflammatory response detectable during TBI may be responsible for the onset of the subsequent neurodegenerative processes. However, new evidence indicates that glial activation may also exert reparative/restorative effects.

Research studies on TBI, however, have identified some of the limitations of the M1 and M2 classification system [44,45], since cells can simultaneously co-express M1 and M2 markers after TBI, thus limiting the M1-M2 paradigm [25,45].

Some investigators have analyzed the M1- and M2 microglia polarization during the acute phase of trauma in detail, demonstrating that both functional states are both activated soon after TBI, but at 7 days post-injury the M2-like phenotype is replaced by M1-like phenotype, expressing high levels of NOX2. The NOX2 inhibition in microglia altering M1-/M2-like balance in favor of the M2-like phenotype significantly reduced oxidative damage in the injured cortex, thus demonstrating that repolarize microglia towards an M2-like phenotype and reduced oxidative damage in neurons was detected [46].

There is in vivo proof that microglia metabolism takes part in the pathophysiology of neurotrauma. In this context, it has been reported that in vivo CX3CL1 (fractalkine) administration reduced ischemic damage [47]. This effect is related to a decreased expression of all genes associated with glycolysis concomitant with a microglial polarization from pro- to anti-inflammatory phenotype. In this context, CX3CL1 induced a metabolic change in microglial cells, increasing the expression of genes related to oxidative phosphorylation and reducing expression of those involved in glycolytic metabolism of glucose, this last strictly associated with the energy production typical feature of the pro-inflammatory phenotype [47]. Another observation is that the microglia ablation does not represent a productive therapeutic strategy since depletion of microglia led to dysregulated neuronal calcium responses, calcium overload and increased neuronal death [48]. Therefore, it would be better to suppress a specific phenotype at a particular time, as reported by Rice et al., showing that while microglia elimination after the lesion led to improved functional recovery, microglia depletion during the insult resulted in greater neuronal loss [49].

PET imaging with translocator protein (TSPO) ligands to visualize microglial activity following TBI in vivo showed that areas with high microglial activation exhibited high levels of brain atrophy even many years after injury. In addition, chronic microglial activation has been detected in regions remote from focal injury, thus reflecting a slowly progressive changes within damaged white matter [50]. 

A recent study showed that disruption of the gene codifying Midkine (MK), a multifunctional cytokine, reduces the lesion extension and improves the neurological deficits after TBI in mice. This improved functional outcome is achieved through the modulation of both neuroinflammatory responses and neuronal apoptosis surrounding the lesion at the early phase after TBI. Furthermore, MK-deficiency suppressed M1 microglia phenotype marker and promoted M2 phenotype counteracting tissue loss [51], thus emphasizing that manipulation of genes involved in microglial polarization status may be a potential therapeutic strategy for TBI.

One interesting study conducted on an experimental TBI rat model reported that secretome of adipose-derived mesenchymal stem dells ameliorated TBI-induced neuroinflammatory environments that caused edema, apoptosis of the neural cells and axonal damage, favoring the upregulation of M2 phenotype while reducing the number of M1 phenotype. Furthermore, in this model, pro-inflammatory cytokine IL-6 and TNF-α levels decreased, whereas anti-inflammatory cytokine TGF-β and TSG-6 resulted significantly increased [52].

Jassam and colleagues in their study described the existence of a biphasic cytokine pattern expression with concurrent proinflammatory (IFN-γ) and anti-inflammatory (IL-4 and IL-10) states at subacute and chronic phases post-TBI [44]. This mixed inflammatory profile reflects the functional diversity of microglia during a complex disease such as TBI, thus this functional overlap suggests that the simplification of the polarization state into the M1/M2 paradigm does not reflect the true diversity of microglia as previously reported [27]. This seems to suggest that the imbalance of the immune responses can promote TBI pathology and exacerbate a chronic neuroinflammatory state.

The failure to resolve the initial powerful pro-inflammatory reaction can invalidate the beneficial effects of the restorative wound-healing immune responses. Hence, it is of crucial importance to know the causes of the inflammatory response, its consequences on degeneration or neuronal survival and how to address or modify differentiated inflammatory reactions in order to drive immune responses favoring reparative processes and restoring CNS function in TBI.

## 3. Extracellular Vesicles (EVs)

### 3.1. Biogenesis of EVs

In CNS, different studies have reported that EVs act as vectors in pathogenesis, intercellular communication, and are used for vaccine or drug delivery, as well as functioning as biomarkers [16,53]. EVs are membranous particles isolable in the main biological fluids of the human body [54,55,56,57,58,59,60,61,62,63,64,65,66,67]. In general, EVs can be divided into three categories, depending on their biogenesis, their mechanisms of release, and their size. Microparticles (MPs)/Microvesicles (MVs) (100–1000 nm) are created by the budding of the plasma membrane; exosomes (Exos) are smaller in size, around 100 nm, and originate from the endosomal/multivesicular body (MVB) system and are accumulated within the cell before their release; and apoptotic bodies (1000–5000 nm) generated from cells during the final phase of apoptotic process. There are two common paths: one involves the endosomal sorting complex required for the transport (ESCRT)-dependent pathway with sphingolipids; the other includes an ESCRT-independent pathway involving tetraspanins. Because of this, the same cell may release different EVs with different cargo with different effects on target cells [68,69]. 

### 3.2. Structure of EVs

Microvesicles and apoptotic bodies show a variable size and shape, instead exosomes display a well-marked round morphology. 

The International Society for Extracellular Vesicles has provided the Minimal Information for Studies of Extracellular Vesicles (MISEV), which was set up in 2014 [70] and then revised in 2018 [71]. MISEV provides guidance for the nomenclature, collection, pre-processing, separation, concentration, characterization, functional studies and reporting. 

According to MISEV, EVs is a generic term for particles that are naturally released from the cell, which are delimited by a lipid bilayer and cannot replicate; in fact, they do not contain a functional nucleus [71].

Besides the MISEV, the EV-TRACK (Transparent Reporting and Centralizing Knowledge in extracellular vesicle research) consortium suggested a tool named EV-METRIC to improve experimental rigor by following a quality chart with nine parameters to increase transparency in the methods used for EV separation and characterization [72].

EVs are composed of a lipid bilayer and contain numerous types of proteins and lipids. The molecular data of EVs are collated in three different databases: (i) Vesiclepedia (http://www.microvesicles.org/) ‘a compendium of molecular data (lipid, RNA, and protein) distinguished in different classes of EVs’ [73]; (ii) ExoCarta (http://www.exocarta.org/) [74], cataloging the contents of Exos from different organisms; and (iii) EV-TRACK (http://evtrack.org/) which use EV-METRIC to create a wide public EV-TRACK knowledgebase from published experiments [72]. Upon endocytic uptake, EVs enter the target cells and bind directly to their plasma membranes or the endosomal membrane. 

Multiple methods and technologies have been developed for the precise detection and isolation of EVs from the biological fluid using their principal properties. The density of the vesicles is an important parameter used to study EVs. Differential ultracentrifugation techniques have been applied to isolate EVs: Exos [75] and exosome-like vesicles [76] with centrifugation at 100,000× *g* or more [77] while MPs/MVs [78] [can be isolated at 10,000× *g* centrifugation [68]. Margolis et al. explain that one strategy for distinguishing between exosomes and the smaller microvesicles is to isolate MVBs and extract their intraluminal exosomes before that they are released, even if the isolation of MVBs is a challenge [79]. 

Flow cytometry, procoagulant assays, and ELISA-based solid phase capture assays [80] are used to characterize EVs, and atomic force microscopy provides structural details [81].

All EVs could be analyzed by electron microscopy and other imaging techniques like Scanning Electron Microscope (SEM) and Transmission electron microscope (TEM) [82]. 

Considering that EVs are separated with different approaches and isolated by different cellular sources, it is still not possible to propose specific and universal markers of one or the other type of EVs, let alone of MVB-derived “exosomes” as compared with other small EVs.

Membrane transport and fusion proteins, tetraspanins, in particular CD9, CD63 and CD81 [83], heat shock proteins, proteins, lipid-related proteins, phospholipases are involved in MVB biogenesis, while surface receptors are the main protein markers of MVs/MPs, integral membrane proteins, as well as cytosolic proteins [84,85].

In apoptotic vesicles there are histones [86], and several studies have founded that histone are also present along DNA in MVs [87,88], while some mRNAs and even miRNAs are present in all types of EVs.

### 3.3. Function and Uptake of EVs

Endocytosis appears to be the primary route of entry for EV [89]; however, they can penetrate cells through other mechanisms, such as clathrin-dependent [90] or -independent endocytosis [91], phagocytosis [92], caveolin-mediated uptake [93], and lipid-raft-mediated absorption [94].

Proteins and glycoproteins found on the surface of EVs may influence their uptake mechanism. [95,96].

Once in the cell, EVs may affect the biological behavior of recipient cells in different ways: EVs could deliver receptors and/or exchange bioactive lipids between cells; EVs may change functional target cells by consigning intracellular proteins or transferring mRNA; acting as signaling complexes, EVs could directly stimulate target cells [97].

It is known that exosomes, MVs, and apoptotic bodies contain miRNAs even if they are generated via independent mechanisms. Different miRNAs have been implicated in neuroinflammation and neurological diseases such as Parkinson’s disease, Alzheimer’s disease, amyotrophic lateral sclerosis, and depression [98,99]. 

MiRNAs transported by EVs are of particular interest as they could influence the gene expression pattern in recipient cells.

Recent studies have addressed the discovery of regulatory mechanisms, whereby cells can selectively control their miRNA cargo of EVs. These mechanisms broadly include RNA-binding proteins such as hnRNPA2B1 and Argonaute-2, but also membranous proteins involved in EV biogenesis such as Caveolin-1 and Neural Sphingomyelinase 2 [100]. 

## 4. EVs in the Brain

In CNS, the communication between glia and neuron is fundamental for different biological functions, such as brain development, homeostasis preservation, neural circuit formation.

Glia cells are not only involved in inflammatory responses during infections or diseases, but they play a key role in remodeling and pruning synapses or as neurotrophic support.

Glia and neurons communicate with direct cell–cell contact, with a paracrine action of secreted molecules, as well as by EVs [101,102,103,104]. EVs are secreted by all brain cells, including neurons [105], astrocytes [106], microglia [107,108], and oligodendrocytes [91,109]. 

EVs could influence neuronal recipient cells in different ways: they may transfer receptors and/or bioactive lipids between cells, by transferring intracellular proteins or transferring mRNA, and stimulating directing target cells [110]. 

The BBB plays an important role in the protection of the brain from neurotoxic substances, preserving its homeostasis. 

It has been demonstrated that BBB dysfunction is associated with numerous neurological disorders, such as multiple sclerosis, stroke, AD and Parkinson’s diseases (PD) [111,112].

Inflammatory stimuli such as TNFα and NF-κB are known to increase BBB permeability and to increase the expression of endothelial adhesion molecules [113].

Chen and colleagues studied the interactions established between Exos and brain microvascular endothelial cells (BMECs), to better understand how Exos could cross the BBB. They have found that Exos transporting luciferase could cross the BMEC monolayer only when cells were inflamed not under normal conditions [114]. 

In a recent study conducted by Dozio et al. in order to understand the existing interaction between EVs and BBB. They firstly isolated and characterized MVs and Exos derived from human brain endothelial cells, and then identified their protein profiles using mass spectrometry (MS)-based shotgun proteomics in control cells and in cells treated with TNFα. They demonstrated that under physiological conditions, MVs contained enriched amounts of mitochondrial and cytoskeletal proteins, whereas Exos enclosed more adhesion, histone and ribosomal proteins involved in exosome biogenesis and cell adhesion. In TNFα-stimulated cells, the phenotype was changed both in MVs and in Exos, resulting in an increasing of several EVs proteins involved in TNF signaling and immune response. Given that EVs may induce changes in phonotype of target cells, EVs released when BBB is inflamed may spread pathophysiological signature to neighboring cells such as astrocytes, pericytes and microglia. Interestingly, the BBB-derived EVs could be present in the circle, so the study of their composition could provide important details on the early processes characterizing neuroinflammatory diseases [115].

Furthermore, EVs are able to act on the BBB phenotype; in fact, Xu et al. demonstrated that in a zebra fish model, Exos containing miR-132 can influence the expression of an adherents junctions (AJ)-related protein (i.e., Cadherin 5 or VE-Cadherin), instead inhibition of miR-132 containing Exos increase BBB permeability and microhemorrhage events in the brain microvasculature [116]. 

The choroid plexus is a site of active protein synthesis that has different receptors for molecules involved in the inflammatory process.

Balusu et al. demonstrated that choroid plexus epithelium (CPE) cells release EVs during systemic inflammation. During inflammation, in fact, CPE cells release EVs in cerebrospinal fluid that contain four inflammatory miRNAs, in particular miR-1a, miR-9, miR146 and miR-155. These CPE-derived miRNA-enriched EVs, travelling in CSF, are taken up by astrocytes and microglia, which in turn amplify the inflammatory response [117].

However, the transport routes employed by these EVs remain obscure, and the impact of EVs on ECs is also unclear.

Recently, Saint-Pol et al. hypothesized the interactions between Exos and endothelial cells in five theoretical ways: (1) association with Gprotein-coupled receptor on the cell surface, inducing a signaling cascade; (2) adhesion to cell surface and fusion, with a release of Exos content in the cytoplasm, which can lead to several types of events, including cell signaling; (3) macropinocytosis; (4) nonspecific/lipid raft; or (5) receptor-mediated transcytosis, leading to its entry into the cell through the endocytic pathway and its storage in the multi vesicular bodies [118].

There is evidence confirming that EVs represent a double-edged sword in CNS diseases: on the one hand, the cells use EVs to remove proteins and toxic aggregates from their cytoplasm; on the other hand, the EVs themselves can directly damage healthy cells, pouring toxic loads into them, contributing to the spread of brain diseases [57].

The release of EVs transporting myelin-associated and other proteins from oligodendrocytes to neurons creates a means of communication between these types of cells and contributes to myelination and neuronal integrity [109,119].

Microglia-derived EVs have been shown to modulate synaptic transmission, by inducing neuronal production of ceramide and sphingosine. The increase of sphingolipid metabolism positively acts on excitatory neurotransmission in vitro and in vivo, reinforcing a physiological modulation of synaptic activity by microglia [120].

Microglia-derived EVs modulate presynaptic transmission via the endocannabinoid system. In fact, microglia-derived EVs transport the endocannabinoid N-arachidonoylethanolamine (AEA) on their membrane and AEA-bearing EVs act on type-1 cannabinoid receptors (CB1) present on GABAergic neurons and inhibit presynaptic transmission [121].

Extracellular ATP is one of the major stimuli for microglia to shed extracellular vesicles through the P2X7 receptors. Different studies have reported that when microglia are stimulated with ATP, they release EVs with IL-1b and GAPDH that promote the propagation and regulation of neuroinflammatory response in the brain [122,123].

Other studies have also reported that the release of microglia-derived vesicles loaded with P2X7 receptors could be an efficient strategy for microglia to decrease the presence of P2X7 receptors on the plasma membrane, decreasing the apoptosis induced by P2X7v [124].

Drago et al. showed that ATP was able to modify the content of MVs derived from microglia driving the synthesis of proteins involved in cellular adhesion to the extracellular matrix, modifying the autophagolysosomal pathway and cellular metabolism, which in turn modulate the cellular response of the receiving astrocytes [125].

In CNS, recent findings attribute to EVs the role of potential carriers in the intercellular delivery of misfolded proteins linked to neurodegenerative disorders, for example Tau and Amyloid b (Ab) in AD, a synuclein in PD, SOD1 in amyotrophic lateral sclerosis, and Huntingtin in Huntington’s disease [126,127,128,129,130]. This makes intriguing the study of EVs that may be potential biomarkers for different chronic neurodegenerative diseases.

## 5. The Emerging Role of Exosomal miRNA in the Diagnosis and Pathophysiology of TBI

As discussed above, TBI is a major public health problem and represents a significant cause of long-term disability and death worldwide in people of all ages [131,132]. TBI complications can cause a wide range of functional changes that affect physical, cognitive, emotional, behavioral and sensorimotor functions, leaving patients with significant post-traumatic disabilities. A large body of evidence has shown that TBI involves a complex interaction of neuropathophysiological mechanisms. Despite recent advances in understanding the cascade of events governing TBI pathophysiology, the underlying mechanisms remain to be fully elucidated [133]. To date, there are no biological tools for detecting mild TBI or to monitor brain recovery. The significant necessity of new diagnostic approaches for identifying neurological trauma and predicting the risk of neurological deterioration in patients with TBI has led to the consideration of endogenous markers. 

In a cell, EVs could have a double power: on one hand, EVs could be received from the cell as a nano-surrogate of their origin cell; on the other hand, they could be released from the cells to remove toxic proteins or aggregate form the cytoplasm of cells, to send biological message to other cells.

Kumar and colleagues investigated the role of microglia-derived microparticles (MPs) in TBI, and they found that the number of MPs 24 h after injury increased. These MPs originated from microglia, demonstrating that microglia-derived MPs were released in the brain following TBI and reached the circulatory system. Moreover, the injection of microglial MPs isolated from TBI in naïve animals has demonstrated that these MPs loaded with pro-inflammatory molecules such as miR155, IL-1b, and TNF-a were able to transfer the post-traumatic neuroinflammatory phenotype to naïve animals. This suggests that MPs are able to propagate neuroinflammation in TBI [12]. 

As Exos carry an abundance of cell-specific biomolecules and are considered to be the fingerprint of the origin cell that generated them, it is believed that they could serve as potential biomarkers of several diseases. Indeed, recent findings have demonstrated that Exos can be used as potential diagnostic markers of TBI, that these nanosized vesicles have an essential role in the intracellular communication, and that they serve as vehicles for transferring biological material between cells [134,135]. As indicated above, Exos may carry a variety of molecular constituents of their cell of origin, including microRNAs (miRNAs). MicroRNAs are a family of non-coding RNAs of 17–24 nucleotides that regulate the expression of several target genes and a wide range of cellular processes [136,137]. Accumulating data indicate that exosomal miRNAs play an important role in the pathogenesis of multiple diseases and involved in the improvement of the treatment outcomes for several diseases. Thus, several miRNAs have been shown to target key neuropathophysiological pathways involved in TBI and have the potential to be developed into novel therapeutic targets. The investigation of their role represents an exciting opportunity. 

Lei et al. observed that after TBI, the expression level of miR-21-5p in the brain increased [138]. This observation is also confirmed by Harrison and colleagues, who found a reduction of miR-212 expression in EVs from TBI mice, while expression of miR-21, miR-146, miR-7a, and miR-7b was significantly increased. They also found that the expression of miR-21 in the brain was primarily localized in neurons adjacent the lesion site and microglia present near these miR-21-expressing neurons were activated [139]. Interestingly, Han et al. found that miR-21-5p was elevated in neurons after TBI [140]. Recently the same group demonstrated that Exos derived from neurons of TBI with highly expressed miR-21-5p, injected into the mouse induced the polarization of M1 microglia. This confirmed the formation of cyclic cumulative damage model between neurons and microglia in the brain through Exos containing miR-21-5p. It is well documented that subarachnoid hemorrhage results frequently from TBI. Lai and co-authors demonstrated that the circulating exosomal miRNA expression profiles showed distinct pattern differences between subarachnoid hemorrhage patients and healthy individuals [141]. Additionally, the injection of modified Exos was able to deliver miRNAs into the CNS, as previously described [142,143]. This suggests that miR-193b-3p delivery may be achieved using exosome encapsulation. The researchers also found that targeted delivery of miR-193b-3p into the brain following subarachnoid hemorrhage reduced neuroinflammation and attenuated neuronal degeneration by inhibition of the HDAC3/NF-κB signaling pathway. Establishing an accurate diagnosis and clinical management of TBI is currently limited due to the lack of accessible molecular biomarkers reflecting the pathophysiology of this heterogeneous disorder. To address this challenge, Ko et al. developed a diagnostic method to characterize the TBI more comprehensively using the miRNA present in the brain-derived EVs. In addition, the authors demonstrated that brain-derived EVs carrying miRNA could be used to characterize specific TBI states and identify the possible signaling pathway in the human brain after injury. Importantly, they also suggested the possibility of using a panel of biomarkers to define the features of a particular lesion rather than looking for a single marker that alone would not be able to distinguish the complex states of a lesion and recovery with reasonable specificity [140]. Interestingly, it was recently demonstrated that an increased level of miR-124-3p in microglia was able to promote anti-inflammatory microglia Type M2 polarization. Moreover, the increased miR-124-3p in microglial Exos exerted a protective effect of inhibiting neuroinflammation. The exosomal miR-124-3p inhibitory effect is mediated by suppressing the activity of mTOR signaling targeting PDE4B. More interestingly, exosomal miR-124-3p promotes neurite outgrowth after scratch injury on one hand. On the other hand, the increased miR-124-3p level in microglial exosomes has been shown to improve the neurologic outcome and inhibit neuroinflammation in repetitive TBI mice [144]. Thus, inhibition of excessive inflammatory response is essential for improving neurologic outcome after TBI and improves functional recovery. As described above, the primary mediators of TBI-induced neuroinflammation are microglia and astrocytes [145]. In a recent study, the release of astrocytic exosomal miRNAs was investigated in an inflammatory stress model. Interestingly, it was shown that the activated primary human astrocytes with IL-1β were able to secrete a subset of five specific exosomal miRNA that may serve as a biomarkers of CNS inflammation and regulate the neuroinflammatory processes. This suggests the important role of exosomal miRNAs in mediating the cascades of inflammatory signaling in the CNS cells [146]. 

Endogenous neuroinflammation in the TBI plays an essential role in the defense of the CNS from the invasion of pathogens and in the repair of tissue lesions. Although neuroinflammatory response is itself largely responsible for the expansion of secondary brain damage and unfavorable outcomes, post-traumatic inflammation also mediates neuroprotective mechanisms after traumatic injury. The dual role of the neuroinflammatory process has sparked the interest of researchers, who have focused their attention on numerous experimental models and clinical studies in recent years, in order to understand the complexity of mechanisms and molecules involved in regulating the post-traumatic inflammatory response [147].

Currently, it has been found a total of 50 exosomal miRNAs that were differentially expressed following TBI. Of these, 31 were upregulated and 19 were downregulated. This suggests the contribution of these altered miRNAs in the progression of TBI [148]. The expression profile of miRNAs in exosomes from the plasma of peripheral blood in a model of induced-TBI in rats was studied by high-throughput whole transcriptome sequencing and subsequent bioinformatics analysis. The authors demonstrated that plasma exosomal components undergoing change after TBI. The exosomal biological material from plasma can reflect the pathophysiological processes after TBI in addition to representing a diagnostic, thus deserving to be considered as a potential target of TBI therapy.

Taken together, exosomal miRNAs are not only used as biomarkers of TBI diagnosis but could also be considered as potential therapeutic targets for TBI treatment. Furthermore, exosomes could be considered novel nano-sized drug delivery systems.

## 6. Extracellular Vesicles as Therapy in TBI

To develop therapeutic strategies to prevent or improve long-term damage and deficits following TBI, it is important to unravel the pathophysiological cascade of events and determine the exact molecular mechanisms underlying the relationship between acute and chronic TBI pathogenesis. Furthermore, many efforts have been made in the attempt to identify strategies for the treatment of neurodegenerative pathologies by enhancing the availability of bioactive agents efficient for targeting cells. Here, we provide a brief overview of the pathophysiology of TBI, highlighting the molecular mechanisms involved, followed by an update on new targets and related therapeutic agents.

As we have demonstrated previously, it has been recognized that EVs are considered to be major vectors by which cells communicate with each other and exchange various biological messages in various disorders [149,150], and in favor of brain recovery following TBI. This recognition has stimulated the interest of a large number of researchers in using EVs to deliver therapeutic agents using a variety of induced TBI. Previous studies have demonstrated that isolated EVs reduced the adverse effects of TBI in mice model [151,152]. More interestingly, the isolated CD63^+^CD81^+^ EVs from mesenchymal stromal cells were able to reduce the rescue pattern separation and the spatial pattern learning impairments after TBI [153]. Moreover, the authors developed an in vivo assay that showed the efficacy of these cells in neuroinflammation suppression after TBI in mice. 

It is well documented in the literature that mesenchymal stem cell-derived Exos are one of the most popular stem cell-derived exosomes, and that they are widely applied in studies about neurological disorders, including TBI [154]. Indeed, recent evidence has found that the early systemic administration of Bone Marrow Stem Cell-derived exosomes shows a promising therapeutic effect in the early stages of TBI. Therefore, treatment with Exos has demonstrated a neuroprotective effect through functional recovery, reduction of the extent of cortical damage, attenuation of cell apoptosis, reduction of neuroinflammatory responses, including the release of cytokines and the polarization of microglia/macrophages [155]. These neuroprotective effects were mediated by brain-derived neuroprotective factor via miR-216a-5p [156].

It has been shown that the transplantation of multipotent mesenchymal stromal cells (MSC)-derived exosomes in a rat model of TBI has significantly improved the spatial learning, sensorimotor functional recovery and neurovascular recovery. The observed functional recovery effects of exosomes acting through promoting endogenous angiogenesis, neurogenesis and reducing neuroinflammation. This suggests that MSC-derived exosomes may provide a novel exosome cell-free therapy for TBI and possibly other neurological diseases [157].

Previous findings from the literature confirm that autophagy plays an important role in the pathophysiologic process of both clinical and experimental models of TBI [158,159,160,161]. In addition, recent studies have indicated that pathways of autophagy are persistently activated after TBI, which may lead to the deterioration of nerve injury [162,163,164].

Taking into consideration that the M2 phenotype of microglia has anti-inflammatory and neuroprotective properties, a treatment that can direct the microglia activated from the M1 forward the M2 phenotype could be an efficient strategy for the treatment of traumatic neurological disorders [165].

Li and colleagues provided evidence that shifting microglia from the M1 to M2 phenotype might be a new strategy for reducing neuroinflammation and improving motor recovery after brain injury [166].

Toll-like receptors are important molecules for initiating immune responses; TLR-4 is expressed in microglia and plays an important role in CNS diseases.

The TLR-4 signaling pathway plays an essential role in the activation and polarization of microglia, and when it is activated, it exerts detrimental effects on CNS diseases [167,168].

A recent study has demonstrated that exosomes released from hypoxic mesenchymal stem cells could shift microglia from M1 to M2 phenotype transferring miR216a-5p by inhibiting TLR-4/NF-KB and activating PI3K/AKT signaling pathway. This study confirms the importance of exosomes with their cargo of miRNA as a therapy in diseases of CNS [168].

Li D. and co-workers showed that the treatment of cultured HT-22 neurons with repetitive TBI mouse brain extracts increased the expression of miR-21-5p in neuron-derived Exos. The exosomal miR-21-5p produced a protective effect by suppressing autophagy in an in vitro TBI model represented by neurons subjected to scratch injury. MiR-21-5p was able to directly affect the Rab11a 3′UTR region to reduce its translation and further suppress neuronal autophagy mediated by Rab11a [169]. Another study by Li D et al. provided evidence that these effects were achieved mainly via upregulation of exosomal miR-124-3p, and that focal adhesion kinase family interacting protein of 200 kDa (FIP200), which plays a key role in trauma-induced autophagy [170]. 

The neuroprotective effect of miR-124-3p was also investigated in a recent study on repetitive mild traumatic brain injury (rmTBI), an important risk factor that predisposes to long-term disorders such as Alzheimer’s diseases.

In injured brain, the level of miR-124-3p in microglial exosomes was significantly altered in the acute, sub-acute, and chronic phases after rmTBI. Treatment for cultured neurons with miR-124-3p (EXO-124) microglial exosomes attenuated neurodegeneration in repetitive scratch-injured neurons.

The impact of microglial exosomal miR-124-3p is exerted by targeting Rela, an inhibitory transcription factor of ApoE that promotes the b-amyloid proteolytic breakdown; thereby, the inhibition of b-amyloid abnormalities contributes to alleviating neurodegeneration [171].

M2 microglia exosomes can be transmitted horizontally to neurons and exert a neuroprotective outcome during ischemic brain injury via exosomal miR-124 [172].

Among the glia cells in CNS, astrocytes are the most abundant cell type and play a fundamental role in neural circuit function and in maintaining microenvironment stability [173,174].

A lot of previous studies have shown that astrocytes and microglia participate actively in many pathological conditions including brain trauma. Activated astrocytes generate many regulatory factors that furnish negative feedback to activated microglia. 

The communication between astrocytes and microglia was confirmed by a study in which the addition of conditioned media from astrocytes on microglia increased antioxidant and anti-inflammatory responses preventing excessive brain inflammation [11].

In a very recent study Long et al. focused their attention on the role of Exos as messengers from activated astrocytes on microglia, they discovered an intriguing role of miR-873 present on astrocyte-derived exosomes that plays an anti-inflammatory effect on microglia.

Activated astrocytes, present in TBI, release exosomes loaded with miR-873 that exert neuroprotective effects through the regulation of microglia activation [175].

In addition, more than 100 miRNAs were found in these Exos derived from astrocytes. The miR-873a-5p is a highly expressed component in traumatized human brain tissue. Furthermore, miR-873a-5p reduces LPS-induced activation of the microglial M1 phenotype and the related inflammation reaction through a reduced phosphorylation of ERK and NF-κB p65. This leads to an improvement in the degree of neurological severity by reducing brain damage in a rigorously cortically controlled murine model. This indicates that exosomal miRNA derived from activated astrocytes plays a key role in the interaction between astrocyte-microglia. Thus, miR-873a-5p, one of the main astrocyte-derived exosomal miRNAs was able to attenuate microglia-mediated neuroinflammation and improve neurological deficits following TBI by inhibiting the NF-κB signaling pathway, leading to polarization from M1 to M2 of microglia. These findings suggest a potential role for miR-873 present in astrocyte-derived Exos in treating TBI [175], suppressing the NF-κB signaling pathway, a classical inflammatory pathway often associated with neurotoxicity [176], so that miR-873a-5p could transform microglial polarization from M1 to M2 phenotype under TBI conditions, thus suggesting possible strategies for developing new drugs. 

In line with the previous findings exploring the role of exosomal micro RNAs in brain recovery after TBI, research results by Yang Y et al. have demonstrated that Exos have been shown to be useful as tool for delivering miR-124 into the brain based on their nano-size advantages, capacity to transfer microRNA, and ability to cross the BBB. This might be a beneficial strategy for improving the outcome of TBI without vascular obstructive effect or risks of tumor formation. Furthermore, the pivotal role of microglia polarization in hippocampal neurogenesis after TBI was emphasized. In this regard, it has been shown that miR-124 was able to move the microglial phenotype towards M2 state, thus improving both neurogenesis in the hippocampus and functional recovery through inhibition of the signal pathway mediated by TLR4 [177]. This could be a possible strategy for improving the outcome of TBI without obstructive vascular side effects or risk of tumor formation [178]. 

Cell-free therapy exosome-based consists of delivering targeted regulatory genes (miRNAs) to decrease neuroinflammation, enhance multifaceted aspects of neuroplasticity, and to enhance neurofunctional recovery for effective and reparative treatment of several neural injuries following TBI. In addition to the role of miRNAs, further investigation of exosomal proteins is warranted to fully investigate the mechanisms of trophic activities underlying exosome-induced therapeutic effects in TBI. Although Exos provide promising therapeutic effects in the rodent model of TBI, as well as in some patients, further studies are required for clinical translation [179].

Despite the prominent and critical role of exosomal miRNA and EV-derived miRNA in ameliorating TBI outcome (see Figure 1) and modulating the brain recovery using the experimental models and clinical researches, further clinical studies are warranted in order to investigate the efficacy in human brain, limiting the side effects, and improving the targeting mechanisms in the human body.

Figure 1 represents EVs loaded with assorted neuroprotective miRNAs released by different CNS cell types.

## 7. Conclusions

In recent years, the attention of research in the CNS field has been focused on EVs and their roles emerging in both physiological and pathological contexts. Different studies have demonstrated their role in neuron–microglia communication, as well as in communication between glia cells. In particular, EVs are the object of several investigations into their capacity to cross the BBB, and for their role in stopping or promoting neuroinflammation in relation to their cargo, as reported in Figure 2.

Figure 2 presents the neuroinflammatory and neuroprotective effects of EVs in relation to their cargo and their contribution to CNS cell network during TBI. 

This figure was created with Biorender.com.

In this review, we have highlighted the key role of Exos with their miRNA cargo (see Table 1) and their protective potential role in TBI for their capacity to improve functional recovery, reduce cortical lesion volume, attenuate cellular apoptosis, and modulate neuroinflammation.

## Figures and Tables

**Figure 1 biomolecules-10-00901-f001:**
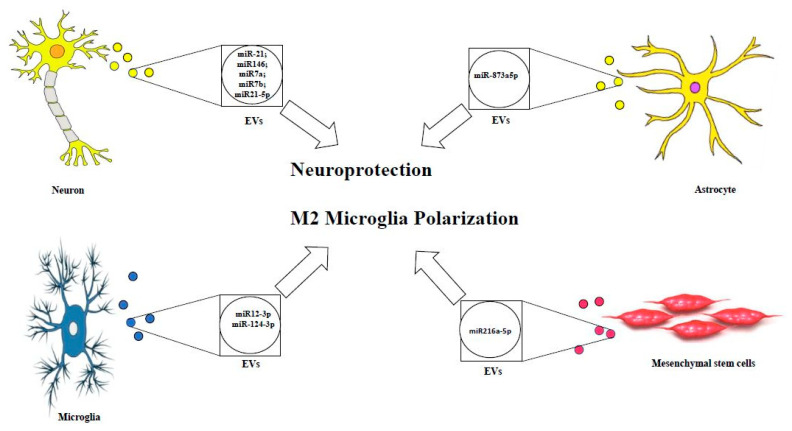
Therapeutic effects of EVs cargo in TBI.

**Figure 2 biomolecules-10-00901-f002:**
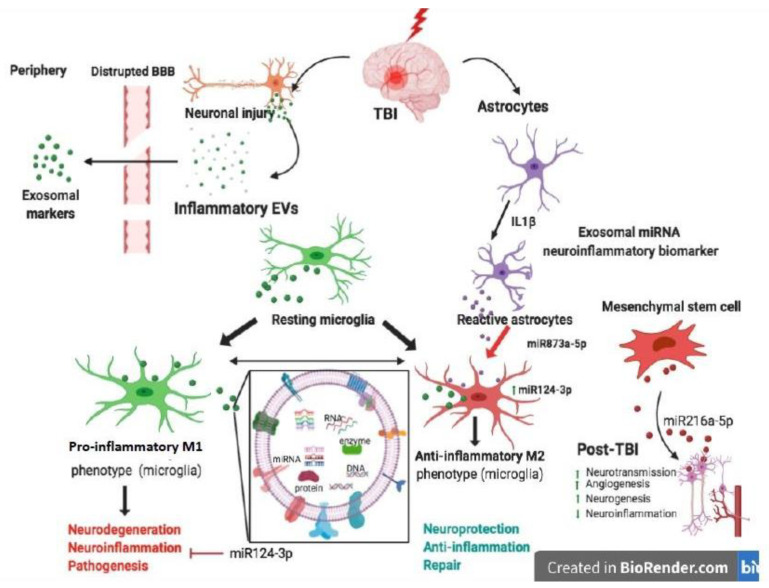
Schematic representation of the effects of EVs in Traumatic Brain Injury.

**Table 1 biomolecules-10-00901-t001:** Immunoregulatory potential and effects of EVs.

Type of EVs	Cell of Origin	Species	Transferring Materials	Biological Effects	References
MPs	Microglia	Mouse	miR155	Inflammation	[11]
MPs and Exos	Neuron	Rat	miR-21; miR146; miR7a; miR7b	Neuroprotection	[139]
Exos	Neuron	Rat	miR21-5p	M1 Microglia Polarization	[180]
Exos	Microglia	Mouse	miR12-3p	M2 Microglia Polarization	[144]
Exos	Mesenchimal stem cell	Rat	miR-216-5p	Neuroprotection; M2 Microglia Polarization	[156,168]
Exos	Neuron	Mice	miR-215p	Neuroprotection	[169]
Exos	Microglia	Mouse	miR-124-3p	Neuroprotection	[170]
Exos	Astrocyte	Human	miR-873a5p	M2 Microglia Polarization	[174]

miRNA-enriched exosomes and MSC-derived exosomes represent emerging biological tools for studying new and effective therapeutic approaches in TBI.

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
