# Peer review of "Extracellular Vesicles miRNA Cargo for Microglia Polarization in Traumatic Brain Injury"

_biomolecules, 2020, doi:10.3390/biom10060901_

Round 1

Reviewer 1 Report

In their manuscript, Panaro et al reviewed the extracellular vesicles miRNA cargo in microglia polarization regards to traumatic brain injury. Over-all, the topic is interesting and timely, though I feel that there are few revisions are warrants.

Introduction is concise and well written.

I would recommend making Section 2 concise as the section seems a large portion of the manuscript.

In section 3. I would suggest the authors discuss the miRNA cargos sorting into exosome and microvesicles.  

Uptake/Internalization of EVs and cargo delivery should be discussed as researchers are focusing on using EVs as a therapeutic agent in TBI as well”.  It would be better to discuss what specific interactions and delivery have been shown in the literature (with figures).

Please refer to the “Identification of a novel mechanism of blood–brain communication during peripheral inflammation via choroid plexus‐derived extracellular vesicles” - EMBO Mol Med (2016)8:1162-1183 for EVs and BBB relationship.

I would suggest adding these publications (PMID: 31244932 and PMID: 31843449) and discuss it in the manuscript. 

Author Response

 We thank the reviewers, which gave us the opportunity, with their suggestions, to improve the quality of the manuscript and add further useful information on the field. We have carefully taken their comments into consideration in preparing our revision, resulting, as we hope, in a paper clearer, more compelling, and broader. Changes in the manuscript are reported in red.

Below are the answers point by point:

Review I

(x) I would not like to sign my review report

( ) I would like to sign my review report

English language and style

( ) Extensive editing of English language and style required

( ) Moderate English changes required

( ) English language and style are fine/minor spell check required

(x) I don't feel qualified to judge about the English language and style

Is the work a significant contribution to the field?  

Is the work well organized and comprehensively described?         

Is the work scientifically sound and not misleading?          

Are there appropriate and adequate references to related and previous work?  

Is the English used correct and readable?   

Comments and Suggestions for Authors

In their manuscript, Panaro et al. reviewed the extracellular vesicles miRNA cargo in microglia polarization regards to traumatic brain injury. Over-all, the topic is interesting and timely, though I feel that there are few revisions are warrants.

Introduction is concise and well written.

I would recommend making Section 2 concise as the section seems a large portion of the manuscript. We thank the reviewer, we reduced the paragraph 2.

In section 3. I would suggest the authors discuss the miRNA cargos sorting into exosome and microvesicles.  We thank the reviewer, we added this information in the paragraph 3.

Uptake/Internalization of EVs and cargo delivery should be discussed as researchers are focusing on using EVs as a therapeutic agent in TBI as well”.  It would be better to discuss what specific interactions and delivery have been shown in the literature (with figures). We thank the reviewer, we added a figure (Therapeutic effect of Extracellular Vesicles cargo in Traumatic Brain Injury

) to the manuscript.

Please refer to the “Identification of a novel mechanism of blood–brain communication during peripheral inflammation via choroid plexus‐derived extracellular vesicles” - EMBO Mol Med (2016)8:1162-1183 for EVs and BBB relationship. We thank the reviewer, we added this reference and we have discussed it.

I would suggest adding these publications (PMID: 31244932 and PMID: 31843449) and discuss it in the manuscript. We thank the reviewer, we have added these references and we have discussed them.

Reviewer 2 Report

This is a description of the role of EV miRNAs in Traumatic brain injury.

I believe there are many areas of improving

  1. it looks like the manuscript has not been extensively revised by authors. there are many typos and spelling errors in the manuscript
    1. abstract line 29 "ore" instead of "are"
    2. page 2  line 78 "are delivery" instead of "are delivered"
    3. page 3 line 105 "toward" instead of "towards"
    4. and many more throughout the manuscript
  2. English is very poor in some parts of the manuscript. Authors should refer to a mother-tongue before resending the manuscript.
  3. Chapter nr.3 on Extracellular vesicles is really poorly written. Authors should gather together information about biogenesis, structure, function, uptake and write them in separate paragraphs. 
  4. On the basis of recent literature it is not correct what authors claim in page 6 line 218-19. Indeed MISEV 2018 states: "ISEV endorses “extracellular vesicle” (EV) as the generic term for particles naturally released from the cell that are delimited by a lipid bilayer and cannot replicate, i.e. do not contain a functional nucleus." Hence it is not correct to use the term exosomes if you are not sure of the biogenesis of EVs.
  5. Another point is that in page 6 authors describe methods and technologies to detect and isolate EVs. This part is poorly written and does not take into account of all the existing methods (e.g TEM is not mentioned although is one of the most used platform for detecting EVs). There are also incorrect statements : authors in line 251 say that by ultracentrifugation at 100000 g one can isolate ectosomes. This is not true, ectosomes are precipitated at lower g. I suggest this review to read: Margolis, L. and Y. Sadovsky (2019). "The biology of extracellular vesicles: The known unknowns." PLoS Biol 17(7): e3000363.
  6. Authors describe also EV markers: they should be more complete and they should not mix protein, RNA but distinguish them.
  7. in general I find this review very long and too full of information without any scheme or figure that could help following the story. Authors should consider to add some figures.

Author Response

We thank the reviewers, which gave us the opportunity, with their suggestions, to improve the quality of the manuscript and add further useful information on the field. We have carefully taken their comments into consideration in preparing our revision, resulting, as we hope, in a paper clearer, more compelling, and broader. Changes in the manuscript are reported in red.

Below are the answers point by point:

Open Review

(x) I would not like to sign my review report

( ) I would like to sign my review report

English language and style

(x) Extensive editing of English language and style required

( ) Moderate English changes required

( ) English language and style are fine/minor spell check required

( ) I don't feel qualified to judge about the English language and style

Is the work a significant contribution to the field?  

Is the work well organized and comprehensively described?         

Is the work scientifically sound and not misleading?          

Are there appropriate and adequate references to related and previous work?  

Is the English used correct and readable?   

Comments and Suggestions for Authors

This is a description of the role of EV miRNAs in Traumatic brain injury.

I believe there are many areas of improving

it looks like the manuscript has not been extensively revised by authors. there are many typos and spelling errors in the manuscript

abstract line 29 "ore" instead of "are"

page 2  line 78 "are delivery" instead of "are delivered"

page 3 line 105 "toward" instead of "towards"

and many more throughout the manuscript

English is very poor in some parts of the manuscript. Authors should refer to a mother-tongue before resending the manuscript.

We thank the reviewer, we have revised and corrected the manuscript.

Chapter nr.3 on Extracellular vesicles is really poorly written. Authors should gather together information about biogenesis, structure, function, uptake and write them in separate paragraphs.

On the basis of recent literature it is not correct what authors claim in page 6 line 218-19. Indeed MISEV 2018 states: "ISEV endorses “extracellular vesicle” (EV) as the generic term for particles naturally released from the cell that are delimited by a lipid bilayer and cannot replicate, i.e. do not contain a functional nucleus." Hence it is not correct to use the term exosomes if you are not sure of the biogenesis of EVs. We thank the reviewer, we have revised the paragraph of Extracellular Vesicles and discussed it more exhaustively in more paragraphs as the reviewer has suggested.

Another point is that in page 6 authors describe methods and technologies to detect and isolate EVs. This part is poorly written and does not take into account of all the existing methods (e.g TEM is not mentioned although is one of the most used platform for detecting EVs). We thank the reviewer, we discussed this part in a concise way because the focus of this review is EVs cargo, we welcome reviewer suggestion and we added this method to the paragraph.

There are also incorrect statements: authors in line 251 say that by ultracentrifugation at 100000 g one can isolate ectosomes. This is not true, ectosomes are precipitated at lower g. I suggest this review to read: Margolis, L. and Y. Sadovsky (2019). "The biology of extracellular vesicles: The known unknowns." PLoS Biol 17(7): e3000363. We thank the reviewer, we deleted ectosomes.

Authors describe also EV markers: they should be more complete and they should not mix protein, RNA but distinguish them. Thank you for this suggestion, we thank the reviewer, we revised this part.

in general I find this review very long and too full of information without any scheme or figure that could help following the story. Authors should consider to add some figures. We have added one figure to the

We added one figure and revised the manuscript according to the suggestions of the reviewers.

Round 2

Reviewer 2 Report

The manuscript has been quite revised according to reviewer suggestion, but there are still some minor revisions to introduce, due to poor english writing in some parts of the work. The reviewer strongly suggest a overall reading of the manuscript by a mother tongue speaker, otherwise the manuscript, which is scientifically sound, will result poorly written in some parts.

Here are my comments:

Page 2 line 42: change  “TBI is pathology multi-phases” in “TBI is a multi phase pathology”

Page 2 line 51: remove “that”, otherwise the sentence is not correct

Page 2 line 67, change “body” to bodies. Change “cells type” in “cell types”

Page 2 line 68, change “produced” in “produce”. Change “EVs content “in “EV content”.

Page 2, line 77, change “between” in “among”

Page 4 line 153 change “ablation not represent” in “ablation does not represent”

Page 5 line change “would seems” “would seem”

Figure 1 is not reported in the revised manuscript. What is the difference with figure 2? Furthermore a caption should be added to the figures.

Author Response

We thank the reviewer, which gave us the opportunity, with their suggestions, to improve the quality of the manuscript and add further useful information on the field. We have carefully taken its comments into consideration in preparing our revision, resulting, as we hope, in a paper clearer, more compelling, and broader. Changes in the manuscript are reported in red.

Below are the answers point by point:

The manuscript has been quite revised according to reviewer suggestion, but there are still some minor revisions to introduce, due to poor english writing in some parts of the work. The reviewer strongly suggest a overall reading of the manuscript by a mother tongue speaker, otherwise the manuscript, which is scientifically sound, will result poorly written in some parts.

Here are my comments:

Page 2 line 42: change  “TBI is pathology multi-phases” in “TBI is a multi phase pathology”

Page 2 line 51: remove “that”, otherwise the sentence is not correct

Page 2 line 67, change “body” to bodies. Change “cells type” in “cell types”

Page 2 line 68, change “produced” in “produce”. Change “EVs content “in “EV content”.

Page 2, line 77, change “between” in “among”

Page 4 line 153 change “ablation not represent” in “ablation does not represent”

Page 5 line change “would seems” “would seem”

We thank the reviewer, we have revised and corrected the manuscript.

Figure 1 is not reported in the revised manuscript. What is the difference with figure 2? Furthermore a caption should be added to the figures.

We thank to the reviewer, the figure 1 was reported at page 16, line 643. The figure 1 represents  the neuroprotection due to EV miRNAs cargo,  the Figure 2 instead represents neuroinflammatory and neuroprotective effects of EVs in relation to their cargo and how EVs establish CNS cells network during TBI.

We have added the captions to the figures.
